# The Importance of Thermally Abnormal Waters for Bioinvasions—A Case Study of *Pistia stratiotes*

Nina Šajna [1],*, Tina Urek [1], Primož Kušar [2] and Mirjana Šipek [1]

1  Faculty of Natural Sciences and Mathematics, University of Maribor, 2000 Maribor, Slovenia
2  Meljski Hrib 30, 2000 Maribor, Slovenia
*  Correspondence: nina.sajna@um.si; Tel.: +386-22-293-705

**Abstract:** Thermally abnormal waters represent safe sites for alien invasive plants requiring warmer conditions than provided by the ambient temperatures in the temperate zone. Therefore, such safe sites are frequently inhabited by tropical and sub-tropical plants. By performing a literature review we assessed that at least 55 alien aquatic plant taxa from 21 families were found in thermally abnormal waters in Europe. The majority of these taxa are submerged or rooted macrophytes. Six taxa are listed as quarantine pests according to EPPO. Among these, *Pistia stratiotes* is present in seven European countries, most of the records of this presence being recent. We studied *P. stratiotes* in a thermally abnormal stream where a persistent population was able to survive harsh winters. Models showed that the optimum temperature for *P. stratiotes* biomass was 28.8 ± 3.5 °C. Here, we show that air temperatures had a higher influence on the photosynthetic efficiency of *P. stratiotes*, estimated by chlorophyll fluorescence measurements, than did water temperatures. Generally, growth, and consequently surface cover for free-floating plants, cannot be explained solely by thermally abnormal water temperatures. We conclude that even though the majority of thermophile alien plant occurrences resulted from deliberate introductions, thermally abnormal waters pose an invasion risk for further deliberate, accidental, or spontaneous spread, which might be more likely for free-floating macrophytes.

**Keywords:** macrophytes; alien invasive plants; chlorophyll fluorescence; plant mass; temperature gradient

## 1. Introduction

Freshwater ecosystems cover less than 1% of the Earth's surface but support around 10% of all known species, making them biodiversity hotspots [1]. However, they are one of the most threatened ecosystems on the global scale, mainly because of land-use change, hydrological system alterations, pollution, climate change, and invasive species establishment [2,3].

Biological invasions of European freshwater ecosystems by alien invasive species have a significant impact on biodiversity and ecosystem functions and cause economic damage [4,5]. However, a small share of freshwater alien plants has a higher economic and ecological impact in comparison to terrestrial plant species [4]. One of the most severe impacts of alien aquatic plants is reducing biodiversity because of rapid vegetative propagation and the ability to outcompete native flora [6,7].

The number of alien aquatic plants in Europe has doubled since 1980 and is still increasing [8]. Hussner [9] reports 96 alien freshwater plant species found in European countries. The majority of them originate from temperate regions similar to Europe (Northern America and Asia), while a smaller share is native to warmer parts of the world (e.g., sub-tropical and tropical regions: Africa, Australia, and Southern America; [9]). The latter group consists of frost-sensitive plants. Low temperatures during the coldest months prevent their establishment, and in general only ephemeral populations are formed during

the warmer part of the year [10]. However, thermally abnormal waters represent an opportunity for frost-sensitive macrophytes to establish and survive cold winters [6,9,11]. Therefore, systems with heated waters, either natural (e.g., geothermal waters) or anthropogenic (e.g., thermal pollution), represent safe sites, which imitate tropical conditions [12]. Consequently, alien macrophyte communities in thermally abnormal waters differ from those in freshwaters with local temperature regimes [13]. However, in light of global warming, we can expect that at least some thermophilous alien macrophytes will increase their introduced distribution range beyond the present restriction to thermally abnormal waters [14]. In this regard, thermally abnormal waters might represent safe sites for potential sleeper invasive plants.

In the present study, we aim to assess a diversity of alien macrophytes recorded in European thermally abnormal waters by performing a literature overview. Later, we focus on a globally invasive pan-tropical macrophyte, *Pistia stratiotes* L. The species can survive winter conditions in thermally abnormal waters [11] and can occasionally occur outside safe sites with thermally abnormal waters during favorable climatic conditions [10]. What is more, a substantial increase in new occurrences in Europe reported recently suggests that deliberate releases might not be the only reason for the range expansion of *P. stratiotes*. Since climate warming can promote the expansion of aquatic plants [15], this might also be the case for *P. stratiotes* [11,16]. Therefore, *P. stratiotes* represents a suitable thermophile macrophyte for investigating the effects of temperature as the most challenging climate condition for tropical plants in the temperate zone. In the thermal backwater in Slovenia, where *P. stratiotes* successfully survived winter for several years, we had an opportunity to study the responses of plants to stress along a temperature gradient that included temperatures that were both low and too high for *P. stratiotes* survival. We aimed to define the most suitable temperature range by studying the year-round surface cover, biomass, and quantum yield of photosynthesis of *P. stratiotes*, using chlorophyll fluorescence.

## 2. Materials and Methods

### 2.1. Literature Review

The list of alien macrophytes occurring in European thermally abnormal waters was made by reviewing the existing literature in Google Scholar and Web of Science. We used the following combinations of words for alien aquatic plants and thermal inland waters: "alien" OR "exotic" OR "invasive" OR "non-native" AND "thermal water" OR" thermally polluted water" OR "thermally abnormal water" OR "heated water" OR "warm water". We further searched specifically for all records relating to *P. stratiotes* and referring to Europe to obtain a list of countries where the species occurs.

### 2.2. Study Species

*Pistia stratiotes* L. (water lettuce) is a herbaceous perennial aquatic macrophyte. Its hairy leaves form free-floating rosettes. It is a clonal plant with stolons that form new rosettes, which eventually detach from the mother plant and start new colonies on their own. Clonal growth is extremely successful, and plants can very rapidly overgrow an entire water surface and form dense mats [17]. *Pistia stratiotes* is found in tropical areas around the world, and its origin is uncertain, though it is likely to be from South America [17]. The species has increased its distribution throughout the sub-tropical regions. It was recorded first in South Africa [18,19] and later also in North Africa [19]. It has a wide distribution in Asia, and it is recorded as invasive [19]. The same is true for the Northern Territory of Australia and for the USA, where it occurs in several states [19]. Its rapid growth alters aquatic ecosystems by decreasing oxygen and photosynthesis in the water underneath mats [20,21]. It clogs waterways, hinders navigation, and damages fisheries. It also enables the transmission in the tropics of certain human diseases spread by mosquitoes and snails. In many countries, it is known as one of the worst pantropical aquatic weeds [22].

## 2.3. Study Site

Our study site is a natural thermally abnormal backwater of the larger Sava River named Topla struga in Prilipe near the village Čatež (45°53′ N, 15°37′ E; Figure 1). A naturalized *P. stratiotes* population was established there in 2001 and persisted until its eradication by local authorities in the spring of 2022. The backwater is a 4 km long stream beginning at a geothermal spring and flowing toward the Sava River. An additional source of warm water is provided by a discharge of the thermal spa, which joins the stream after passing through a water treatment plant (Figure 1B). The temperature of the spring water is above 30 °C and cools downstream. The situation at the study site represents a permanent continuously decreasing temperature gradient all year round. We positioned 10 permanent sampling points along the stream for season-long continuous monitoring (Figure 1B). The first five sampling points were 100 m apart and the last five were 200 m apart. Data for the water temperatures of the Sava River at the Jesenice na Dolenjskem recording location (5 km downstream of the study site) and the air temperatures at the Novo Mesto meteorological station were obtained from the national Slovenian Environment Agency (ARSO) [23]. For comparisons, we used temperature data from known locations with *P. stratiotes* distribution in Corumba, Brazil (https://en.tutiempo.net/records/sbcr) (accessed on 7 November 2018) [24] and Eldoret, Kenya (https://en.tutiempo.net/records/hkel) (accessed on 7 November 2018) [25].

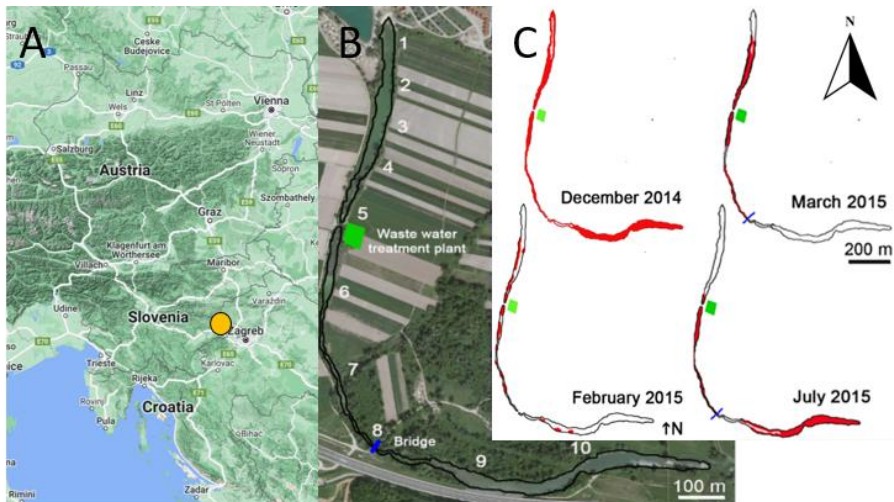

**Figure 1.** (**A**) Study site in Slovenia shown by the orange circle; (**B**) ten permanent sampling points along the thermal stream Topla (the green square indicates the location of the wastewater treatment plant); and (**C**) surface cover of *P. stratiotes* from December 2014 to July 2015 marked with color red.

## 2.4. Sampling Design

### 2.4.1. Abiotic Parameters

The following abiotic conditions were measured weekly between March and July 2015. Measurements took place at some point between 9 a.m. and 2 p.m. At each sampling point, the following were measured:

Air temperature and air humidity 1.5 m above the water surface, using an aspiration psychrometer (Ahlborn FNAD 46, Holzkirchen, Germany);

- Photosynthetically active radiation (PAR) on the water surface, using a quantum sensor (Quantum Meter; Apogee Instruments, Roseville, CA, USA);
- Water temperature at 10 cm depth, using a digital thermometer (HI 98501; Hanna Instruments, Woonsocket, RI, USA);
- Water temperature of the surface, using an infrared thermometer (572-2, Fluke, Washington, DC, USA).

Simultaneously, water samples were collected to obtain the pH value (HI 98103; Hanna Instruments, Woonsocket, RI, USA), the nitrate concentration (Visocolor®ECO Nitrate; Macherey-Nagel, Düren, Germany), and the phosphate concentration (Visocolor®ECO Phosphate; Macherey-Nagel, Düren, Germany). All data are included in the Supplementary Material (Table S1). Initial multiple regression analysis showed that, among measured temperatures, which significantly predicted fresh biomass and leaf length, air temperatures showed the highest correlation with beta ($\beta = 0.53$, $p < 0.001$) and were therefore included in models as well as water temperatures.

### 2.4.2. Seasonal Surface Cover

We monitored the surface cover of *P. stratiotes* in the stream Topla from December 2014 to July 2015. We marked distributional data for December, February, March, and July on the map and digitized it using ArcView GIS 3.2. [26].

### 2.4.3. Ecophysiological Measurements In Situ

Five plants were sampled randomly at each sampling point for the following ecophysiological measurements (Table S2). The photochemical activity of the plant was detected by measuring chlorophyll *a* (Chl-*a*) fluorescence of photosystem II (PS II) using a portable fluorescence spectrometer (Handy PEA; Hansatech Instruments, Norfolk, UK) following the manufacturer's instructions. We recorded the $F_v/F_m$ value representing the maximum quantum yield of PS II, which is highly correlated with the quantum yield of net photosynthesis, whereby $F_v$ is the variable fluorescence ($F_m–F_o$) and $F_m$ is the maximal fluorescence value [27]. $F_v/F_m$ in healthy plants is consistently around 0.83. The ratio $F_v/F_m$ is a measure of stress, if smaller than 0.8, especially if the value falls after dark-adaptation measurement compared to ambient measurement. Chlorophyll fluorescence measurements were performed immediately and after a 10 min period of dark adaptation to obtain actual and dark-adapted fluorescence values, respectively. The dark-adaptation time of 10 min was determined experimentally according to the device manual. Fluorescence measurements were taken on healthy leaves without signs of damage to exclude the effect of stress.

### 2.4.4. Biomass Studies

After ecophysiological measurements in the field, plants were collected and stored in plastic bags in a refrigerator until measurements were taken of the rosette diameter and of the longest leaf length (Table S2). Because of the strong correlation between these two variables (Pearson's r = 0.97), only leaf length was included in the generalized linear models (GLMs). The plants' fresh and dry biomass (after drying at 60 °C for 72 h) were weighed.

### 2.5. Data Analyses

In the observed range, we had sites with no living specimens on both sides of the temperature gradient. To extract the water temperature range for *P. stratiotes* survival in our site, we used nonlinear regression models for modeling the temperature dependence of plant growth, which included the Gaussian functions of fresh plant mass and leaf length. Nonlinear parameter estimates were obtained using the nonlinear least squares method. We used zero values for cases where there was zero abundance. For the chlorophyll fluorescence data, we used GLMs to analyze the influence of the air temperature and sampling date (both as continuous variables) during the observation period. We applied GLMs to the actual and dark-adapted values of the fluorescence data.

## 3. Results

### 3.1. Aquatic Plants Alien to Europe Associated with Thermally Abnormal Waters

European thermally abnormal waters include natural thermal waters, discharges of thermal spas, industrially heated waters from power plants, and mining. Countries in which aquatic plants alien to Europe were recorded in thermally abnormal waters include

Austria, Germany, Hungary, Poland, Romania, Serbia, Slovakia, Slovenia, Ukraine, the European part of Russia, and Iceland, while no reports from other European countries were found (Appendix A, Table A1).

Altogether, 55 alien aquatic plant taxa from 21 families were found, each one having been observed in thermally abnormal waters in at least one location. Among them, six taxa are included in the EPPO A2 List (Appendix A, Table A1) [28]. The alien aquatic plants found comprised the following growth forms: 18% free-floating (10 taxa, two of them ferns), 13% leaf floating (7 taxa), 33% submersed (18 taxa), and 36% emersed (18 taxa, one of them a fern).

The most numerous were the Hydrocharitaceae family (8 taxa), followed by Nymphaeaceae (6), Pontederiaceae (5), Plantaginaceae (4), Onagraceae (4), Araceae (4), Acanthaceae (3), Haloragaceae (3) and Alismataceae (3). The remaining 12 families had one or two alien representatives (Appendix A, Table A1). The vast majority of alien aquatic plants associated with thermally abnormal waters originate from subtropical and tropical climates.

The most common alien aquatic plant in warm waters are *Pistia stratiotes* and *Vallisneria spiralis*, both found in seven European countries, followed by *Egeria densa*, which was recorded in six countries. *Cabomba caroliniana*, *Hygrophila polysperma*, *Monochoria korsakowii*, and *Shinnersia rivularis* were found in the warm waters of four European countries. Of the 55 alien aquatic plant taxa, 60% (33 taxa) were found in only one country, the majority of them in Hungarian thermal waters (Appendix A, Table A1).

*3.2. The Current Distribution Range of Tropical Macrophyte P. stratiotes in Europe*

The literature review assessed the occurrence of *P. stratiotes* in 21 European countries (Figure 2). The first record is dated 1966 [29]. In some locations, introductions happened several times. For example, in Germany, the latest introductions resulted in persistent populations (see discussion for the Erft River in Germany). In some countries *P. stratiotes* was recorded in the past but is no longer present. In Spain, it was immediately eradicated after it was found [30]. In Croatia, it was found once in the summer and completely died during the following winter. Occurrences in the 1980s and 1990s (Table 1; Figure 2) were recorded as ephemeral. Later records report increasing occurrences outside of thermally abnormal waters.

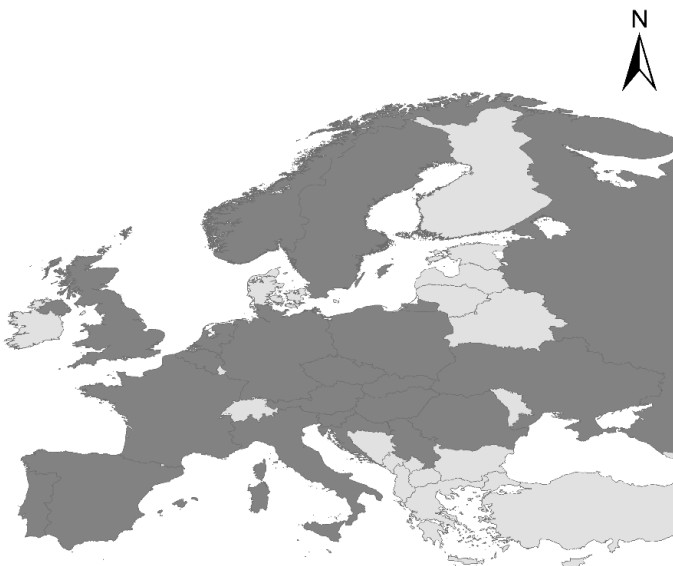

**Figure 2.** European countries where *Pistia stratiotes* has already been recorded are in grey. For details see Table 1.

**Table 1.** Distribution of *P. stratiotes* within European countries and year of the first report.

| Country | The First Report (Location) | Reference |
|---|---|---|
| Austria | 1980 | [31] |
| Belgium | 2000 | [32] |
| Croatia | 2017 (River Sava) | [10] |
| Czech Republic | 1991 | [33] |
| France | 1992 (Auvergne) | [34] |
| Germany | 1981 (thermally polluted River Erft) | [35] |
| Hungary | 1966 | [29] |
| Italy | 1998 (Bodrio le Margherite) | [36] |
| Netherlands | 1973 | [37] |
| Norway | 1989 (Lake Stilla, Ak Skedsmo) | [38] |
| Poland | 2012 (Lake Stawiki in Sosnowiecu) | [39] |
| Portugal | 1990 | [40] |
| Romania | 2005 | [41] |
| Russia | 1989 (River Volga) | [42] |
| Serbia | 1994 (thermal spring Banjica) | [43] |
| Slovakia | 2007 | [44] |
| Slovenia | 2001 (thermal stream Topla) | [11] |
| Spain | 2001 (Guipúzcoa) | [30] |
| Sweden | 2006 | [45] |
| United Kingdom | 1983 (pond in London) | [46] |
| Ukraine | 2011 (River Seversky Donets) | [47] |

*3.3. A Case Study from Thermal Stream Topla*

3.3.1. Physical and Chemical Parameters of the Environment

The air humidity during the sampling period was between 40% and 64%, with an average of 53%. Photosynthetically active radiation varied at between 365 and 1249 μmol/m$^2$s (average 765 μmol/m$^2$s). The highest values were measured in June and July (Table S1).

The pH values varied at between 7.4 and 8.9 (average 8.1). The concentration of nitrates in the water was between 0.50 and 3.75 mg/L, with the highest values measured in April. The average nitrate content was 1.83 mg/L. The concentrations of phosphates in the water varied at between 0 and 0.43 mg/L (average value of 0.15 mg/L; Table S1).

3.3.2. Thermal Conditions

The study site is located in a typical temperate climate, characterized by a harsh winter lasting from 21st December until 21st March with temperatures well below 0 °C and snowfall. At the nearest meteorological station, the lowest average monthly air temperatures for 2015 were measured in January and February, −1.7 and −1.6 °C, respectively. In 2015, the lowest daily maximum of −15.4 °C was recorded in January. Summer lasts from June 21st until September 23rd. Temperatures can rise above 30 °C. In 2015 the highest monthly average temperatures were recorded in July and exceeded 29 °C, with a daily maximum of 36.2 °C [23]. Comparison with the temperatures in the tropics, where *P. stratiotes* is native (Corumba, Brazil), and in the subtropics, where *P. stratiotes* is invasive (Eldoret, Kenya), shows that from the beginning of April until the end of September the air temperatures in our study site were as high as in the tropics and subtropics (Figure 3).

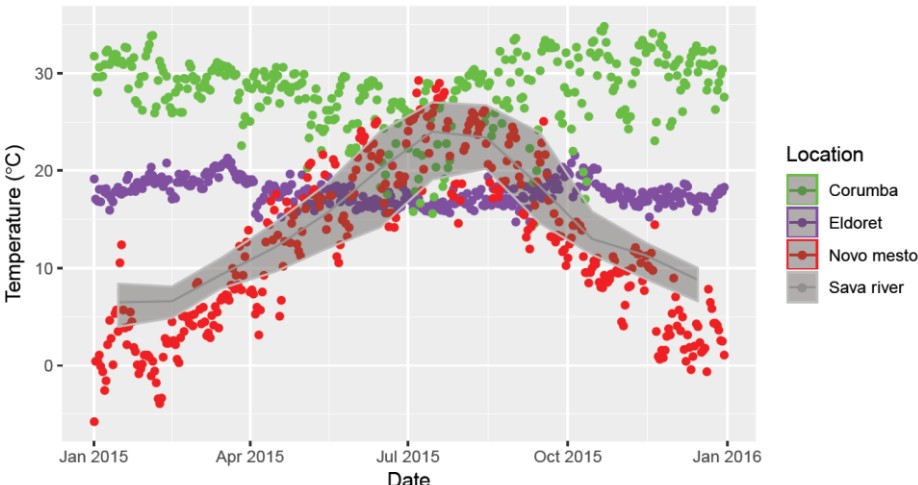

**Figure 3.** Comparisons of daily mean air temperatures in 2015 at the Novo Mesto meteorological station near Čatež (Slovenia), in Corumba (Brazil), and in Eldoret, (Kenya). Monthly mean water temperatures at the Jesenice na Dolenjskem recording site on the Sava River, 5 km downstream of the study site (Prilipe, Slovenia), obtained from the Slovenian Environment Agency are shown by the dark grey area: average temperature (dark grey line) and minimum and maximum temperatures (light grey lines).

The temperatures of the Sava River 5 km downstream of the study site (Figure 3) were strongly correlated with the air temperatures; however, they never fell below 6 °C. The highest average water temperatures were in July (24 °C) and August 2015 (23 °C), with the warmest day peaks reaching as high as 29 °C.

In our study site, the mean water temperature at the source was 27 °C in March and reached 33 °C in May, June, and July (Figure 4, Table S1). The last and also the coolest sampling point on our gradient had a mean temperature in March of about 15 °C and increased with the season to a mean of 27 °C in July. With the warming season, the water temperatures increased with time, resulting in the shift of temperature gradient downstream. In addition, the mean daytime air temperatures increased from 10 to 30 °C during our observation period (Table S1). Water temperatures varied less in time; however, they had a clear site dependency. Sites closer to the thermal source had 10 to 15 °C higher temperatures (Figure 4, Table S1), a temperature difference that persisted during our observations.

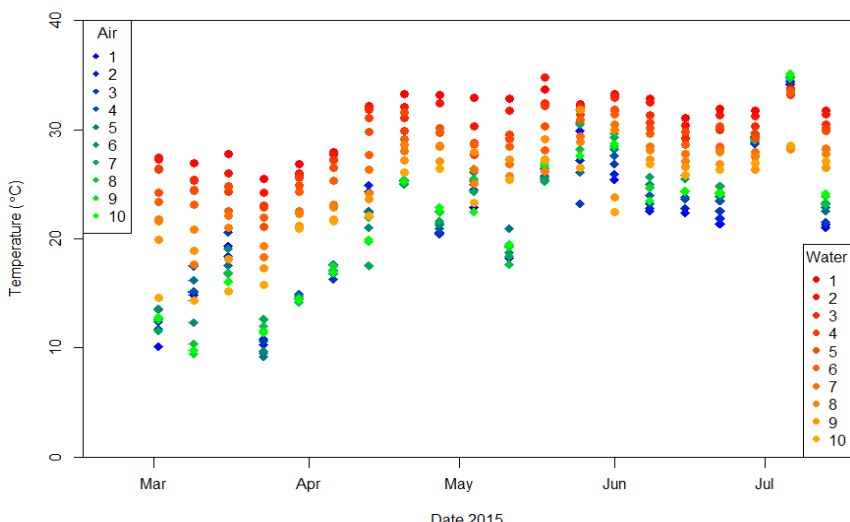

**Figure 4.** Air temperature 1.5 m above the water surface and water temperature at 10 cm depth at 10 permanent sampling points along thermal stream Topla measured weekly from March to July 2015.

### 3.3.3. Seasonal Surface Cover

The cover and spread of *P. stratiotes* were greatest at the end of the growing season in December, which was then followed by die-offs caused by winter (Figure 1C). Plants survived in the upper part of the stream and started to increase their cover again in March and April, occupying the stream at sampling points 1 to 8. In May, plants were distributed along all 10 sampling points on the gradient. In June and July, plants were missing at the warmest locations (sampling points 1 to 3). In the winter it was too cold at the end of the stream, while in the summer, with the increase in the air temperature, locations at the hot source were too hot for *P. stratiotes* (Figure 5B).

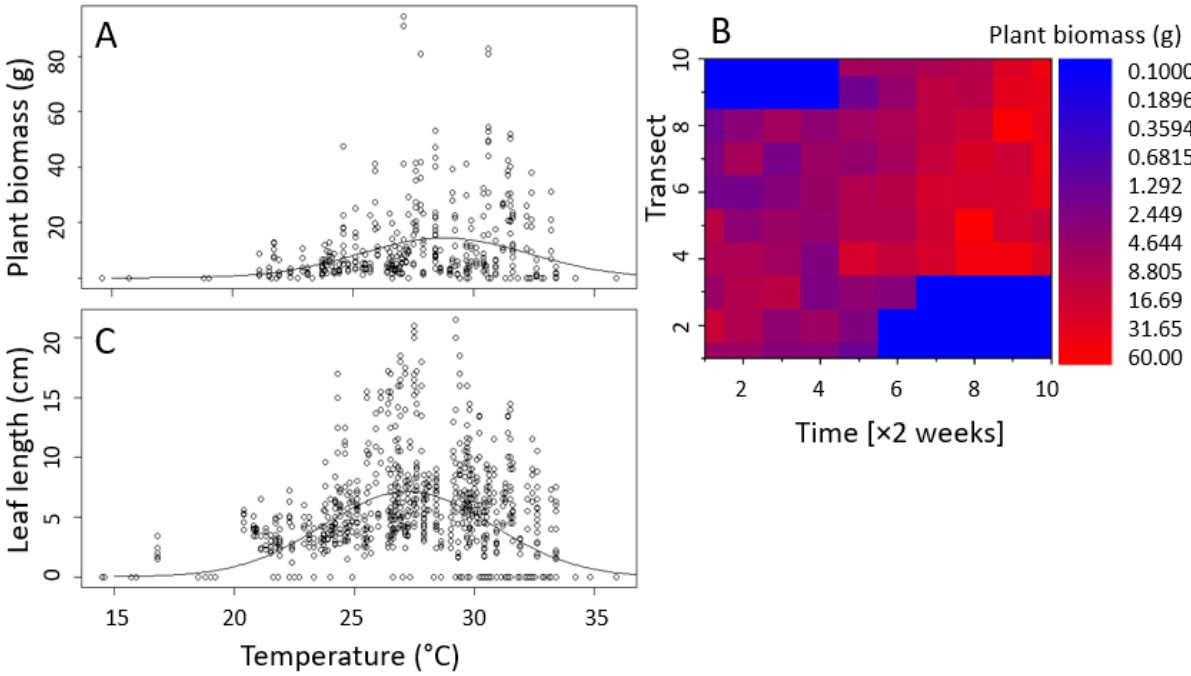

**Figure 5.** Effect of water temperature on fresh *Pistia stratiotes* biomass (**A**,**B**) and leaf length (**C**).

### 3.3.4. Effect of Temperature on Biomass and Chlorophyll Fluorescence

Water temperatures at which plant biomass reached the highest values were slightly higher than temperatures at which we observed plants with larger leaves (Figure 5A,C). The nonlinear least-squares predictions were centered at 28.8 °C, with 1 sigma of 3.5 °C for fresh plant mass and 27.2 °C with the same sigma for the leaf length (Table 2).

**Table 2.** Estimated regression parameters, standard errors, *t*-values, and *p*-values for the models explaining the effect of water temperature on fresh plant mass and leaf length of *Pistia stratiotes*. Model: $x \sim a \times \exp(-(T - T_0)^2/(2 \times \text{sigma}^2))$.

|  | Estimate | Std. Error | *t*-Value | *p*-Value |
|---|---|---|---|---|
| Fresh mass |  |  |  |  |
| a | 14.372 | 1.034 | 13.900 | <0.0001 |
| $T_0$ | 28.779 | 0.324 | 88.944 | <0.0001 |
| sigma | 3.516 | 0.402 | 8.744 | <0.0001 |
| Leaf length |  |  |  |  |
| a | 7.131 | 0.211 | 33.80 | <0.0001 |
| $T_0$ | 27.231 | 0.133 | 204.16 | <0.0001 |
| sigma | 3.516 | 0.158 | 22.26 | <0.0001 |

The photochemical efficiency, estimated by measuring actual chlorophyll fluorescence, increased with the season. For the first few months (from March until May), the actual chlorophyll fluorescence ratio $F_v/F_m$ was below 0.6 for the entire gradient (Figure 6A). Chlorophyll fluorescence measured after a 10-minute-long dark adaptation, during which reaction centers of photosystem II (PSII) were able to return to a relaxed state, showed some increase. However, the values never exceeded 0.75 in March, April, and May (Figure 6B). In June and July, the actual as well as the dark-adapted $F_v/F_m$ increased (Figure 6), particularly at the last two sampling points of the gradient. The difference between actual and dark-adapted values was the smallest in July. At the same time, at more than half of the sampling points, plants had $F_v/F_m$ above 0.8 on average. The outcome of the GLM shows that actual chlorophyll fluorescence was significantly dependent on the sampling date and air temperature (Table 3).

**Table 3.** Estimated regression parameters, standard errors, *t*-values, and *p*-values for the actual chlorophyll fluorescence and after-dark adaptation of *Pistia stratiotes* leaves.

|  | Estimate | Std. Error | *t*-Value | *p*-Value |
|---|---|---|---|---|
| Chl Fluorescence $F_v/F_m$—actual (GLM, Gaussian) |  |  |  |  |
| Intercept | 0.464 | 0.020 | 22.873 | <0.0001 |
| Sampling date | 0.019 | 0.001 | 14.213 | <0.0001 |
| Temperature | −0.004 | 0.001 | −3.509 | <0.001 |
| Chl Fluorescence $F_v/F_m$—dark adapted; $F_v/F_m$ (GLM, Gaussian) |  |  |  |  |
| Intercept | 0.702 | 0.015 | 45.845 | <0.0001 |
| Sampling date | 0.009 | 0.001 | 8.729 | <0.0001 |
| Temperature | −0.002 | 0.001 | −2.198 | 0.028 |

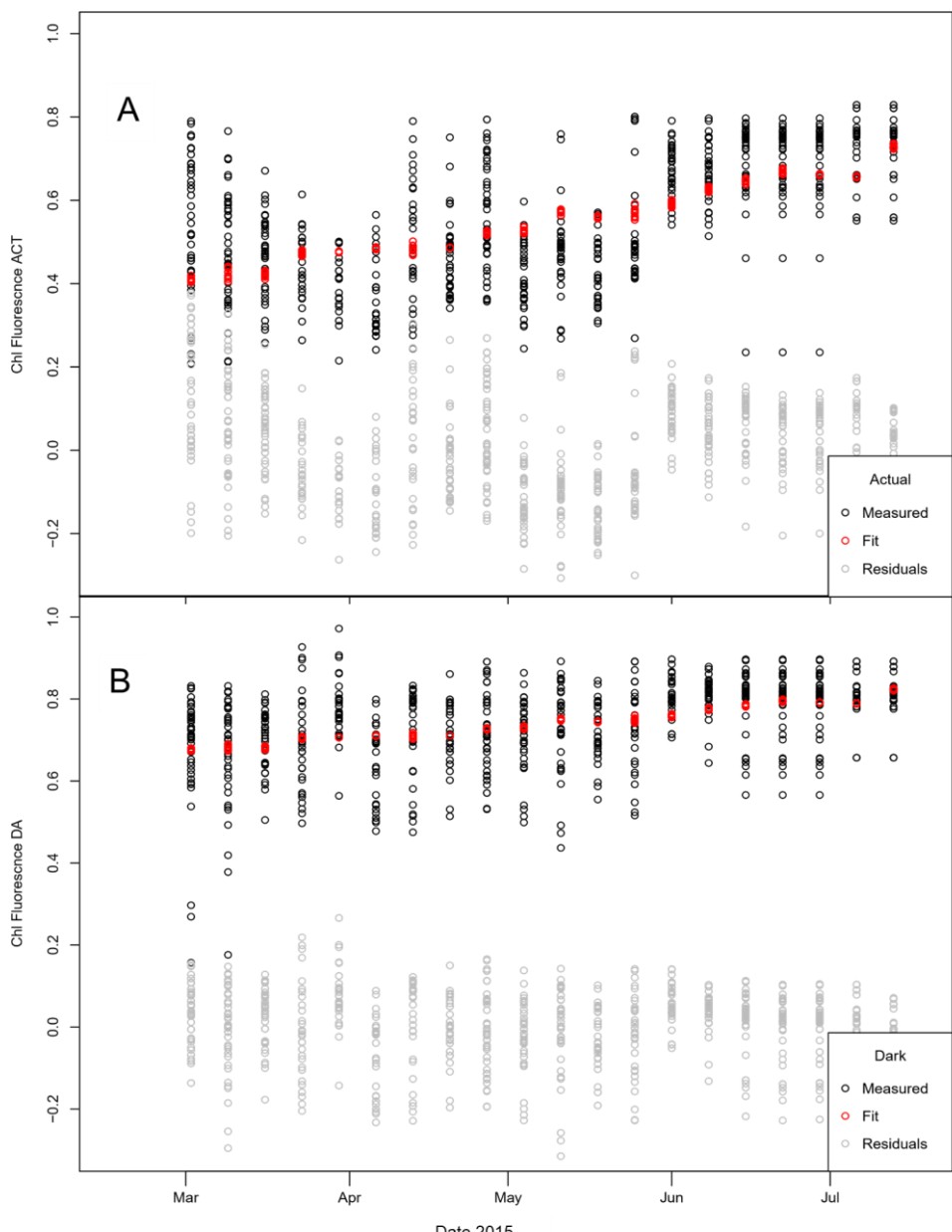

**Figure 6.** (**A**) Actual and (**B**) dark-adapted chlorophyll *a* fluorescence measured as $F_v/F_m$.

## 4. Discussion

### 4.1. Macrophytes in Thermally Abnormal European Waters

Fifty-five alien macrophytes were associated with thermally abnormal waters in Europe, which corresponds to about half of all alien macrophyte species recorded in Europe [9]. Lukács et al. [29] reported that, in Hungary, 80% of alien macrophytes were restricted to thermal waters. However, the frequency of these macrophytes was relatively low, pointing to their invasive potential being restricted by habitat availability, when compared to alien macrophytes that prefer cold waters.

The vast majority of alien macrophytes found in thermal waters originate from deliberate releases (93%), i.e., they are used as ornamental plants either in aquaria or outdoors (e.g., pond plants). Therefore, emersed or leaf-floating macrophytes with prominent flowers (e.g., *Nymphaea*) and/or free-floating macrophytes with large leaves (e.g., *Pistia stratiotes*) are overrepresented in the literature assessed. The significant role of aquarists and the trade in ornamental alien macrophytes in freshwater-ecosystem invasions is already well

recognized and addressed in several studies [29,48,49]. The group of alien plants likely to have originated from aquaria was mainly represented by fast-growing submerged plants (e.g., *Vallisneria*). Alien plants introduced unintentionally were rare and included small free-floating plants (e.g., *Wolffia globosa*) or submerged plants (e.g., *Myriophyllum heterophyllum*). It is worth mentioning that, in European thermally abnormal waters, two of the most problematic aquatic plants worldwide were present: *Eichhornia crassipes* and *Pistia stratiotes*.

Only a smaller proportion of macrophytes found in thermally abnormal waters originate from climates similar to Europe, while approximately 80% of them have natural distribution in subtropical and tropical areas. For alien macrophytes originating from the temperate zone, European thermally abnormal waters are a habitat where the growing season is extended, while their survival is also possible at ambient conditions. For the invasive species that are native to warmer conditions, the current distribution range in Europe is likely to expand in the Mediterranean and Atlantic regions, and possibly also in continental parts, because of milder winters indicative of global warming. Further expansion in all regions can be expected because of increasing waterbodies changes: thermal pollution, channeling, urban spread incorporating waterbodies, gravel pits, shallow artificial lakes, or ponds [11,16,50]. Shallow low-velocity aquatic habitats might be particularly suitable for ephemeral colonization during summers. What is more, analysis of the field data for the Netherlands' ditches showed that milder winters enhanced eutrophication by increasing the total phosphorus concentrations, and favored evergreen submerged and, especially, free-floating macrophytes—the latter because of their ability to obtain nutrients from the water column [51,52]. This is of concern, given that some alien macrophytes recorded have a reputation as extremely problematic and costly weeds in subtropical regions (e.g., *Pistia stratiotes* and *Eichhornia crassipes*) [53]). According to our study, locations reported in the literature often contained several co-occurring invasive alien plants. One of the reasons—besides deliberate introduction—might be that the invaded habitat was altered by invasive macrophytes in a way that favors the establishment of other invasive species more than native species [54].

### 4.2. Pistia Stratiotes Is Expanding Introduced Distribution

We assessed *Pistia stratiotes* to be the most recorded alien invasive plant in thermally abnormal waters. Although the data could be biased, since the species is easily recognized, and smaller or submerged macrophytes can be easily overlooked, there is no doubt that *Pistia stratiotes* is a popular aquatic macrophyte for aquaria and outdoors. Its mass cultivation is increasing the probability of irresponsible releases into nature, which provide the main opportunity for ephemeral population establishment and naturalization possibilities across Europe. Especially worrying is the recent increase in records of established populations, and of populations consistently present for a longer time, in the southern part of Europe.

It is generally accepted that the first introduction to Europe happened in 1973 in the Netherlands [37]; however, there is also an earlier notation from Hungary in 1966 [29]. Early introductions are believed to have been deliberate releases of aquaria plants, especially since several of the first recordings were from ponds (e.g., 1983 in London, UK [46]); thermal springs (e.g., 1994 in Banjica, Serbia [43]), or thermally polluted waters (e.g., 1981 in Erft, Germany [6] and 1989 in Astrakhan, Russia [42]). Ephemeral occurrences sometimes resulted in expansions during summers in the 1980s and 1990s [55,56]. In 2001, the first overwintering of *Pistia stratiotes* in the temperate zone with winter temperatures well below 0 °C was recorded for a population in a thermal water location [11]. After that, the number of ephemeral observations in the temperate zone increased, including those outside thermally abnormal waters. In some locations, *P. stratiotes* was successfully eradicated (e.g., in Spain [30]), while in some countries the number of observations increased considerably after the year 2000 (Figure 7), e.g., from 49 to 200 in the Netherlands [57] and to 36 in the UK. In Somerset (UK), besides more frequent sightings, beginning with several records in 2004, the populations changed also in their abundance. In 2010, *P. stratiotes* was already well established in the Bridgewater and Taunton Canal [58]. In six European

countries with the most records of *P. stratiotes*, 86% of all observations were recorded after 2012 (Figure 7) [45]. However, we have to consider that a high number of records does not necessarily reflect a wide occurrence of the species. Several records in the Global Biodiversity Information Facility (GBIF) [45] database include observations from wildlife recording applications (e.g., iNaturalist or Pl@ntNet), which allow multiple inputs of the same species at the same locality. Nevertheless, such data give information about the actual distribution of a species and the persistence of its populations.

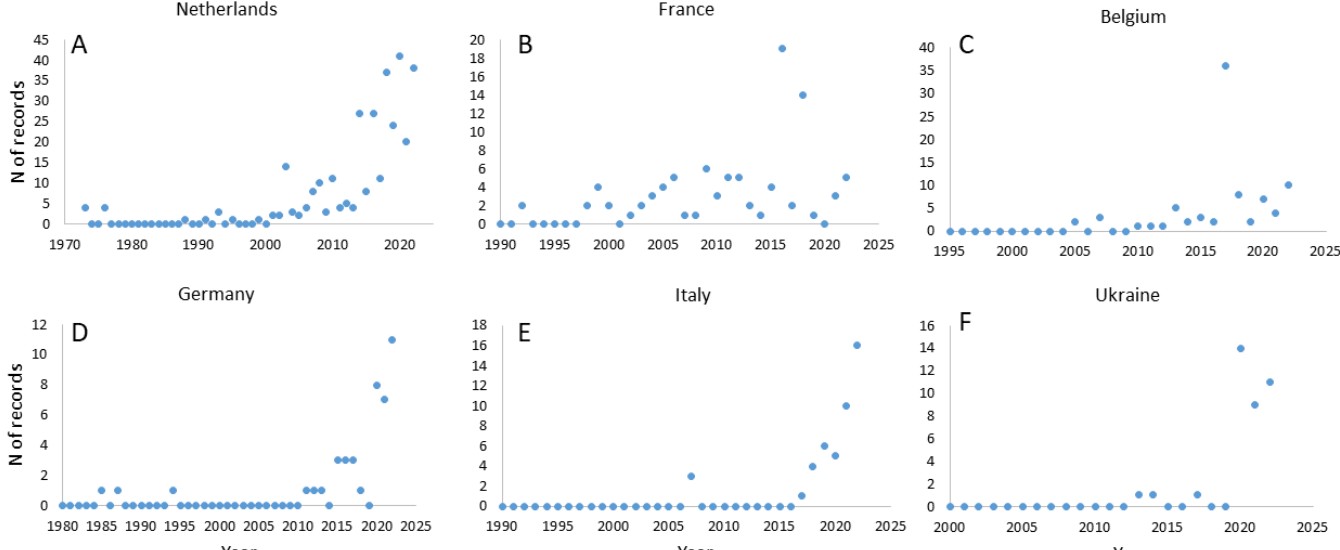

**Figure 7.** Increase in *Pistia stratiotes* records in the database GBIF since the year 1980 for selected European countries: (**A**) the Netherlands, (**B**) France, (**C**) Belgium, (**D**) Germany, (**E**) Italy, (**F**) Ukraine.

The majority of recent recordings remain ephemeral occurrences in normal waters during summers. However, in the last two decades, several established populations have been recorded in various habitats. Since its introduction, *Pistia stratiotes* has remained present in natural thermally abnormal waters in Hungary [29], and it was present until 2022 in Slovenia. Since 2008, it has been established in the thermally polluted River Erft in Germany [6], where it was previously introduced in 1981, but failed to survive [35]. Persisting populations are found in the Mediterranean area. According to the database SI Observation Flore [34], water lettuce was observed in France as early as 1992, in Auvergne. Two more locations were recorded in 1998 and 1999. After that, it occurred abundantly in the River Moselle between 2002 and 2015. In summer, it has been spotted in 31 locations throughout France, with dense, persisting populations in the canals along the River Rhone [59]. *Pistia stratiotes* was also recorded in Italy, where mass reproduction was reported in 1998 in Lombardy [36] and in 2007 in Toscana [60]. Recent distribution ranges include, additionally, the Campania, Emilia-Romagna, and Veneto regions [16,61]. In Spain, *P. stratiotes* was first observed in the province of Gipuzkoa in 2001. In 2004, a population was recorded in the vicinity of Doñana National Park in Andalusia [30]. The information was communicated rapidly to the environmental authorities, and the plant was removed from the continental part of Spain according to the EPPO [19], although it is invasive on the Canary Islands. In Portugal, the first reported naturalization of water lettuce was in 1990 [40].

The mild climate has also enabled persistently recurring populations for more than five years in the South of England and Belgium (since 2000) [32] and Slovakia (since 2007) [44,62]. Increases in population numbers have been reported from less favorable climatic regions as well, e.g., 15 populations in the Czech Republic since the first observation in 1999 [33], while mass reproduction was recorded in 2013 in Ukraine (near Ekshar) [47]. Persisting populations have been reported from the European part of Russia [63]. Habitats invaded by persistent *P. stratiotes* populations in Europe are very similar and are characterized

by slow-flowing, shallow waters, often in canals, experiencing strong human influence. Plants in natural waters form smaller rosettes with smaller leaves than plants in thermally abnormal waters [62].

Additionally, new records became more frequent in regions with a more pronounced temperate climate. *P. stratiotes* was found in 2005 in Romania [41] and in 2012 in Poland [39], and in 2017 it was observed for the first time in a natural river in Vojvodina, Serbia, near the Romanian border [64]. It was also recorded as far north as Sweden (in 2006 and 2011) [45] and Norway, where it managed to form a dense population during the summer of 1989. However, it did not survive winters [19,38].

In several locations, thermally abnormal waters pose an invasion risk for further *P. stratiotes* spread. Spontaneous *P. stratiotes* spread from a thermal safe site was recorded for the first time in 2017, when plants were flushed from the thermal stream Topla in Slovenia into the Sava River and managed to colonize a natural reserve area in Croatia—the Sava-Strmec, 15 km downstream—during the summer [10]. A similar spread can be expected from the thermally polluted Erft River in Germany, because of the high number of plants and seeds drifting downstream into the River Rhine [65,66].

Similar to EU countries, the increase in *P. stratiotes* records in Japan was noted around the year 2000. Until then, *P. stratiotes* had been present, since the 1930s, only on the Okinawa Islands, while the expansion to many sites, including an overwintering population in thermally abnormal water in Japan's temperate zone, began after the late 1990s ([67] and references therein). This might be an indication that additional *P. stratiotes* expansion is occurring globally. Additionally, the spread of *P. stratiotes* in Europe recently might be attributed to global warming [16,17], which might also be the case globally because the spread of *P. stratiotes* into subtropical and tropical regions is enhanced [68].

### 4.3. Effect of Temperature on P. stratiotes Performance

Temperature and photoperiod are abiotic variables that directly influence *P. stratiotes* [69]. This is shown in our study. Even though in our study site winter air temperatures fell below 0 °C, plants survived in the warmer part of the gradient. The suitability of the thermal gradient for *P. stratiotes* growth changed during the season, since parts close to the thermal source became too warm in June and July, while the lower part warmed up as summer approached and turned out to be the most suitable for high photosynthetic yield from May onwards. The wider tolerance of free-floating freshwater macrophytes for low sub-optimal temperatures than for high sub-optimal temperatures [70] is also confirmed for *P. stratiotes* in our study. However, because one can measure more extreme temperature tolerance values if such temperature is applied for only a short time [71], we need to measure the response of plants which are under long-term exposure or have had the opportunity to adapt to an environment with periods of sub-optimal temperatures. In Brazil and in our study site, similar high water temperatures (24 to 28 °C) were most favorable for *P. stratiotes* growth. These water temperatures were present in Brazil in August [69] and in our study site in July. According to our study's regression models, the water temperatures at which maximum plant biomass was reached were 28.8 ± 3.5 °C.

Such mean temperatures were present in the lower part of the thermal gradient in July, which was also the part of the gradient that was closest and most similar to the natural Sava River. This poses the threat of a spillover of plants [10,65] or their seeds [7,11] from the thermal safe site into natural waters during the warm summer. If we take into consideration the sigma range of 3.5 °C, favorable temperatures above 25.3 °C were already present along the entire gradient in May. However, at that time, plants had a low $F_v/F_m$ ratio, indicating a low quantum yield of PSII, which increased after dark adaptation to 0.72 on average. The dark-adapted fluorescence parameters reflect the degree of acclimation to local environmental conditions, and we can assume that in favorable environmental conditions the plants can express their best potential photosynthetic performance [72]. Therefore, the failure of the dark-adapted $F_v/F_m$ to reach the critical value of 0.8 in May showed that plants were under stress. In June, the mean dark-adapted $F_v/F_m$ reached 0.8

at the very end of our temperature gradient. In July, the mean dark-adapted $F_v/F_m$ was higher than 0.8 for the last three sampling points.

In comparison, the water temperature gradient in May ranged from 33 °C to 25 °C, while the mean air temperature was between 22 and 23 °C. In June and July, the water temperature gradient (from 33 °C to 26 °C) did not change considerably compared to May, while the mean air temperatures rose along the entire gradient to 25–27 °C in July and to 26–28 °C in August. According to our results, we can conclude that increased air temperatures in June and July had an important role in the increased $F_v/F_m$ ratio indicating that plants were not under stress. At the same time, dark-adapted $F_v/F_m$ above 0.8 reflected a high quantum yield of PSII and therefore photosynthetic efficiency, which is shown also by increased plant biomass in June and particularly for the last sampling points in July. The importance of air temperature was also stressed in the tropics, where winter air temperatures (16 °C at rosette height) were unfavorable [69]. According to our results in the temperate zone with pronounced winters, the projected global warming will not enable *P. stratiotes* to overwinter in the form of rosettes outside thermally abnormal waters, because the average winter temperature would need to increase significantly, which is unlikely to happen.

## 5. Conclusions

According to the presented literature review, we can conclude that in thermally abnormal waters submerged thermophile macrophytes have a higher probability of survival. Submerged plants are surrounded by warm water with more or less stable temperatures. On the other hand, free-floating thermophile macrophytes depend also on air temperatures. This is also the reason why almost 50% of non-native macrophytes in European thermally abnormal waters are submerged or rooted, and free-floating macrophytes represent only 18% of species. Here we have shown that we should pay attention to free-floating macrophytes, which can survive in thermally abnormal waters and which, at the same time, can tolerate low air temperatures and even considerable frost damage to their leaves. Additionally, there might be an indication that free-floating plants such as *P. stratiotes* have a slightly higher possibility of dispersal compared to rooted macrophytes. Increased water flow or elevated water levels caused by rain or changed discharges can enable easier spillover of free-floating plants than of rooted, submerged plants. What is more, waterfowl, pet dogs, and water equipment constitute additional possible vectors of dispersal. Free-floating macrophytes originating from warmer regions, particularly those plants that do not have leaves close to the water surface, are very interesting subjects for investigation of their temperature tolerance when air temperatures are limited.

**Supplementary Materials:** The following supporting information can be downloaded at: https://www.mdpi.com/article/10.3390/d15030421/s1, Table S1: Physical and chemical parameters of the environment measured at 10 permanent sampling points along thermal stream Topla in 2015; Table S2: Morphological and ecophysiological characteristics of *P. stratiotes* measured at 10 permanent sampling plots along thermal stream Topla.

**Author Contributions:** Conceptualization, N.Š.; methodology, N.Š. and T.U.; formal analysis, M.Š. and P.K.; investigation, T.U.; data curation, N.Š., M.Š. and P.K.; writing—original draft preparation, N.Š. and M.Š.; writing—review and editing, N.Š., P.K., and M.Š.; visualization, N.Š., M.Š. and P.K.; supervision, N.Š.; project administration, N.Š.; funding acquisition, N.Š. All authors have read and agreed to the published version of the manuscript.

**Funding:** This research was partly performed within the project Creative Path to Practical Knowledge funded by the Slovenian Ministry of Education and Research and partly funded by the Slovenian Research Agency ARRS (program P1-0403 and project J1-2457).

**Institutional Review Board Statement:** Not applicable.

**Data Availability Statement:** The data supporting reported results are included as the supplementary material Tables S1 and S2.

**Acknowledgments:** The authors would like to acknowledge Petra Arh for her help in the field. The authors are grateful to three anonymous reviewers for their insightful comments that improved the final paper.

**Conflicts of Interest:** The authors declare no conflict of interest.

## Appendix A

**Table A1.** List of alien aquatic plant taxa associated with thermally abnormal waters in Europe, including family, pathway, growth form, and EPPO A2 categorization. For each taxon, the year of the first record in a particular country and the native distribution are provided. Source of thermal abnormality (if the information was provided in the reference): G = natural geothermal spring, TP = thermal pollution, D = discharge of a thermal spa.

| Taxon | Introduced Range (Year of the First Observation) | Native Distribution | Reference |
|---|---|---|---|
| *Azolla filiculoides* Lam. (Azollaceae); aquarium plant; free-floating fern | Germany (1991, River Erft—TP) Hungary (1940) | North, Central, and South America [9] | [6,29,73] |
| *Bacopa caroliniana* (Walt.) B.L. Robins (Plantaginaceae); aquarium plant; emersed | Hungary (2005) | Southern parts of North America [74] | [29] |
| *Bacopa monnieri* (L.) Wettst. (Plantaginaceae); aquarium plant; emersed | Hungary (2005) | Australia, Africa, Asia, South America, and southern parts of North America [74] | [29] |
| *Cabomba caroliniana* A. Gray (Cabombaceae); aquarium plant; submersed | Hungary (several localities; 1937, Héviz—G) European Russia (1990, backwater of the Moskva River—TP) Austria (2010 (20. cent.), thermal stream Warmbad Villach—D, G) Romania (2014 Peţa Lake—G) | South America [9] | [29,75–78] |
| *Ceratopteris thalictroides* (L.) Brongn. (Pteridaceae); aquarium plant; emersed fern | Hungary (1968) Austria (2013, thermal stream Warmbad Villach—D, G) Romania (2014, Peţa Lake—G) | Widespread in tropical regions [9] | [29,77,78] |
| *Cryptocoryne crispatula* subsp. *balansae* (Gagnep.) N.Jacobsen (Araceae); aquarium plant; submersed | Austria (2013, thermal stream Warmbad Villach—D, G) | N Vietnam, Thailand [77] | [77] |
| *Cryptocoryne wendtii* de Wit (Araceae); aquarium plant; submersed | Austria (2013, thermal stream Warmbad Villach—D, G) | Thailand [74] | [77] |
| *Egeria densa* Planch. (Hydrocharitaceae); aquarium plant; submersed | Germany (1980, River Erft—TP) Slovakia (1993) Hungary (1960) Russia (1983, Pekhorka River—TP) Iceland (2013, pond in Husavik—G; 2004, Opnur—G) Austria (1910, thermal stream Warmbad Villach—D, G) | South America [9] | [6,29,44,73, 76,77,79] |

**Table A1.** *Cont.*

| Taxon | Introduced Range (Year of the First Observation) | Native Distribution | Reference |
|---|---|---|---|
| *Eichhornia crassipes* (Mart.) Solms (Pontederiaceae); ornamental plant; free-floating; EPPO A2 | Hungary (1950) Germany (2005, River Erft—TP) | South America [9] | [6,29] |
| *Eichhornia diversifolia* (Vahl) Urb. (Pontederiaceae); aquarium plant; free-floating | Hungary (2005) | South America [74] | [29] |
| *Gymnocoronis spilanthoides* DC. (Asteraceae); aquarium and aquatic ornamental plant; emersed; EPPO A2 | Hungary (1988, Lake Héviz—G) | South America [74] | [29] |
| *Heteranthera zosterifolia* Mart. (Pontederiaceae); aquarium plant; submersed | Serbia (2016, Niška Banja—G) Austria (2013, thermal stream Warmbad Villach—D, G) | South America [74] | [77,80] |
| *Houttuynia cordata* Thumb. (Saururaceae); ornamental plant; emersed | Hungary (2005) | Southeast Asia [74] | [29] |
| *Hydrilla verticillata* (L. f.) Royle (Hydrocharitaceae); aquarium plant; submersed | Hungary (1980) Slovakia (1995) Austria (1907, thermal stream Warmbad Villach—D, G) | Asia [44] | [29,44,77] |
| *Hydrocotyle ranunculoides* Lf. (Apiaceae); ornamental pond plant; free-floating; EPPO A2 | Germany (2003, River Erft—TP) Hungary (2005) | North, Central, and South America [9] | [6,29] |
| *Hygrophila corymbosa* Lindau. (Acanthaceae); aquarium plant; emersed | Hungary (2005) | Southeast Asia [9] | [29] |
| *Hygrophila difformis* Blume (Acanthaceae); aquarium plant; emersed | Hungary (2005, thermal pond and parts of the Danube near the Lukács Budapest—G) | S Asia and SE Asia [9] | [29,81] |
| *Hygrophila polysperma* (Roxb.) T. Anderson (Acanthaceae); aquarium plant; emersed | Hungary (1958, thermal pond and parts of the Danube near the Lukács Budapest—G) Germany (2005, River Erft—TP) Poland (2008, lakes of the Konin Valley area—TP) Austria (2005, thermal stream Warmbad Villach—D, G) | India and Malaysia [9] | [6,29,77,81, 82] |
| *Lagarosiphon major* (Ridl.) Moss (Hydrocharitaceae); aquarium plant; submersed | Hungary (2005) Austria (1966, thermal stream Warmbad Villach—D, G) | S Africa [9] | [29,77] |
| *Lemna minuta* Kunth (Lemnaceae); unintentional introduction; free-floating | Germany (1981, River Erft—TP) Serbia (2016) Russia (2008, Pekhorka River—TP) | North, Central, and South America [9] | [6,73,76,80] |
| *Lemna aequinoctialis* Welw. (Lemnaceae); unintentional introduction; free-floating | Germany (1982, River Erft—TP) | E Asia, southern hemisphere [74] | [35] |
| *Limnophila heterophylla* (Roxb.) Benth. (Plantaginaceae); aquarium plant; submersed | Romania (2014, Peţa Lake—G) Hungary (2018, thermal pond and parts of the Danube near the Lukács, Budapest—G) | India, SE China, Philippines [74] | [78,81] |

**Table A1.** *Cont.*

| Taxon | Introduced Range (Year of the First Observation) | Native Distribution | Reference |
|---|---|---|---|
| *Limnophila sessiliflora* Blume (Plantaginaceae); aquarium plant; submersed | Hungary (1940) Slovakia (1993, Bojnice) | Asia [44] | [29,44] |
| *Ludwigia* sp.—hybrid (Onagraceae); aquarium plant; emersed | Hungary (2018, thermal pond and parts of the Danube near the Lukács, Budapest—G) | / | [81] |
| *Ludwigia alternifolia* L. (Onagraceae); ornamental pond plant; emersed | Hungary (1940) | E North America [74] | [29] |
| *Ludwigia grandiflora* (Michx.) Greuter & Burdet (Onagraceae); ornamental pond plant; emersed; EPPO A2 | Hungary (2005) | South America, parts of North America [74] | [29] |
| *Ludwigia repens* J.R. Forst. (Onagraceae); aquarium plant; emersed | Hungary (1924) Serbia (2011, Niška Banja—G) Slovakia (2017) | North and Central America [74] | [29,44,83] |
| *Mimulus guttatus* Fisch. ex DC. (Scrophulariaceae); ornamental pond plant; emersed | Hungary (1994) | North America [74] | [29] |
| *Monochoria korsakowii* Regel & Maack (Pontederiaceae); ornamental pond plant; emersed | Hungary (1988) | Ukraine, Caucasus, India, East Asia [74] | [29] |
| *Myriophyllum aquaticum* (Vell.) Verdc. (Haloragaceae); aquarium plant; submersed | Hungary (1968) Germany (2003, River Erft—TP) Romania (2014, Peţa Lake—G) Austria (1988, thermal stream Warmbad Villach—G, D) | South America [74] | [6,29,73,77, 78] |
| *Myriophyllum heterophyllum* Michx. (Haloragaceae); aquarium plant, ornamental pond plant; submersed; EPPO A2 | Hungary (2006) | E-North America, Central America [74] | [29] |
| *Myriophyllum tuberculatum* Roxb. (Haloragaceae); aquarium plant; submersed | Hungary (2018, thermal pond and parts of the Danube near the Lukács, Budapest—G) | India to China, N Peninsula Malaysia [74] | [81] |
| *Najas gracillima* (A. Braun ex Engelm.) Magnus (Hydrocharitaceae); unintentional introduction; submersed | Hungary (2012) | Australasia, China, Eastern Asia, North America [74] | [29] |
| *Najas guadalupensis* (Spreng.) Magnus (Hydrocharitaceae); aquarium plant; submersed | Slovakia (1986) Hungary (2005) Romania (2014, Peţa Lake—G) | America [74] | [29,44,78] |
| *Nelumbo nucifera* Gaertn. (Nelumbonaceae); ornamental pond plant; leaf floating | Hungary (1955) | Ukraine to north Iran, Russian Far East to Tropical Asia, Australia [74] | [29] |
| *Nuphar advena* (Aiton) W.T. Aiton (Nymphaeaceae); ornamental pond plant; leaf floating | Hungary (1920) | North America [74] | [29] |
| Nymphaea "Blue Bird" (*N. micrantha* x *N. capensis*) (Nymphaeaceae); ornamental pond plant; leaf floating | Hungary (1900) | / | [29] |
| *Nymphaea lotus* var. *thermalis* L. (Nymphaeaceae); ornamental pond plant; leaf floating | Hungary (1842) | Endemic to the thermal water of the Peţa River in Romania [84] | [29] |

**Table A1.** *Cont.*

| Taxon | Introduced Range (Year of the First Observation) | Native Distribution | Reference |
|---|---|---|---|
| *Nymphaea nouchali* var. *caerulea* (Savigny) Verdc. (Nymphaeaceae); ornamental pond plant; leaf floating | Hungary (1891, thermal pond and parts of the Danube near the Lukács, Budapest—G) | E Africa [74] | [29,81] |
| *Nymphaea rubra* Roxb. ex Andrews (Nymphaeaceae); ornamental pond plant; leaf floating | Hungary (1891) | Tropical Asia [74] | [29] |
| *Pistia stratiotes* L. (Araceae); aquarium plant, ornamental pond plant; free-floating; EPPO A2 | Hungary (1966) Germany (1981, River Erft—TP) Russia (1989, Volga—TP; 1998 Pekhorka River—TP) Serbia (1994, a thermal spring "Banjica"—G; 2005, Rgoška Banja spa—G) Slovenia (2001, stream Topla—G, D) Slovakia (2007) Ukraine (2011, Seversky Donets—TP) | Pantropical distribution, probably originating from South America [17] | [6,11,29,35, 42,44,47,76, 80] |
| *Pontederia cordata* L. (Pontederiaceae); ornamental pond plant; emersed | Hungary (2005) | North America, South America [74] | [29] |
| *Rotala rotundifolia* (Buch.-Ham. ex Roxb.) Koehne (Lythraceae); ornamental pond plant; emersed | Hungary (1998) Serbia (2016, Niška Banja—G) | Tropical Asia [74] | [29,80] |
| *Sagittaria latifolia* Willd. (Alismataceae); aquarium plant; emersed | Austria (1951, thermal stream Warmbad Villach—D, G | North America, Central America, N South America [74] | [77] |
| *Sagittaria platyphylla* (Engelm.) J.G. Sm. (Alismataceae); aquarium plant, ornamental pond plant; emersed | Russia (2002, Pekhorka River—TP) | S North America, Central America [74] | [76] |
| *Sagittaria subulata* (L.) Buchenau (Alismataceae); aquarium plant; submersed | Hungary (1965) Slovakia (1995) Germany (1984, Warme Wuhle—TP) | SE North America, N South America [74] | [29,44,85] |
| *Salvinia auriculata* Aubl. (Salviniaceae); aquarium plant; free-floating fern | Hungary (1964) | Central and South America [74] | [29] |
| *Saururus cernuus* L. (Saururaceae); ornamental plant; emersed | Hungary (2005) | E North America [74] | [29] |
| *Shinnersia rivularis* (A.Gray) R.M.King & H.Rob. (Asteraceae); aquarium plant; emersed | Germany (1992, River Erft—TP$_m$) Slovakia (1998, 2002, a waste canal discharging thermal water from the bath house "Kalinka", Bojnice—D, G) Hungary (1998, a thermal lake Hévíz—G) Austria (2000, thermal streams in Warmbad Villach—G) | Central America [9] | [29,44,86] |
| *Utricularia gibba* L. (Lentibulariaceae); aquarium plant; free-floating | Hungary (1936) Slovakia (1993) | America, Africa, Asia [74] | [29,44] |
| *Vallisneria americana* Michx. (Hydrocharitaceae); aquarium plant; submersed | Russia (2010, Pekhorka River—TP) | North America, Central America, N South America [74] | [76] |

**Table A1.** *Cont.*

| Taxon | Introduced Range (Year of the First Observation) | Native Distribution | Reference |
|---|---|---|---|
| *Vallisneria gigantea* Graebn. (Hydrocharitaceae); aquarium plant; submersed | Hungary (1891) | SE Asia, Australia [74] | [29] |
| *Vallisneria spiralis* L. (Hydrocharitaceae); aquarium plant; submersed | Hungary (1808)<br>Russia (1999, Desnogorsk Reservoir—TP; 1972, Belovskoe Reservoir—TP)<br>Germany (2003, River Erft—TP; 2017, Reden—TP)<br>Serbia (2011, Niška Banja—G)<br>Poland (1993, Konin Lakes—TP)<br>Iceland (2013, pond Husavik—G)<br>Austria (1880, thermal streams in Warmbad Villach—D, G) | N Africa, Asia, S Europe [9] | [6,29,73,77, 79,83,87– 91] |
| *Victoria amazonica* Sowerby (Nymphaeaceae); ornamental pond plant; leaf floating | Slovakia (1998) | South America [74] | [44] |
| *Wolffia globosa* (Roxb.) Hartog & Plas (Araceae); unintentional introduction; free-floating | Russia (2002, Pekhorka River—TP) | Pakistan to Japan, Malaysia [74] | [76] |

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
