# Peer review of "The Importance of Thermally Abnormal Waters for Bioinvasions—A Case Study of Pistia stratiotes"

_diversity, doi:10.3390/d15030421_

Round 1

Reviewer 1 Report

My opinion is that this paper is suitable for publication.

I would like to point out the following:

In the chapter Introduction, the authors cited references that indicate that they know the field of research and that helped them in choosing a scientifically interesting topic.

Materials and methods are clearly described and logically presented. Field research is conducted with a clearly defined experiment design and the stated procedures necessary to test the set hypothesis and obtain relevant own results. Statistical methods are appropriate to the research topic and objectives.

The results are clearly divided into chapters in accordance with the objectives, clearly presented and logically commented. The results are presented in an appropriate and logical sequence in tables and graphs and further clarified in the discussion.

The discussion logically follows the sequence used in presenting the results.

At the end of the paper, the conclusions are given in which a summary of the obtained results is given, with an emphasis on the new scientific contribution.

Author Response

Dear reviewer thank you very much for your time and comments. Here we list the major changes we made:

- Table 1 has moved to Appendix 1

- former Appendix 1 is now added as Supplementary material

- The order in Results is changed: now we report about the literature review first

 - we added additional literature and improved the English language

The authors

Reviewer 2 Report

The study written by Nina Šajna and colleagues focuses on Pistia stratiotes, alien aquatic plant, their occurrence in thermal water and ecological, physiological and biomass characteristics. The manuscript could be interesting for research of the international journal Diversity. However, according to my opinion the manuscript in recent form is a hybrid between review and traditional research papers. Part of the manuscript is focused on review about occurrence of the aquatic plants in thermal waters in Europe and especially, P. stratiotes. Second part showed results of ecological and physiological research of P. stratiotes in specific thermal water in Slovenia done by the authors. I think that the manuscript should be divided into two single manuscripts. The first, review about alien aquatic plants of thermal waters in Europe and the second, results of the research of P. stratiotes in Slovenia. However, such a review paper would need much more work than a research paper.

Therefore, the manuscript should be re-written and subsequently again submitted into Diversity journal, but only its “research” part; review paper needs greater work input.

Introduction could be changed and more focused on targed species, P. stratiotes. The review about P. stratiotes in Europe, now presented in the Results chapter, could be redirected to Methods chapter (e.g. within subchapter “Studied species”) and Discussion chapter. Review part about other alien aquatic plants should be omitted, similarly as the part of the Discussion chapter which focuses on the European alien aquatic plants.

Please see also paper „Ružičková, J., Lehotská, B., Takáčová, A. et al. Morphometry of alien species Pistia stratiotes L. in natural conditions of the Slovak Republic. Biologia 75, 1–10 (2020). https://doi.org/10.2478/s11756-019-00345-5“.

Author Response

Dear reviewer thank you very much for your time,  literature update, and helpful comments improving our paper. Here we list the major changes we made first and later address all the comments in detail:

- Table 1 has moved to Appendix 1

- former Appendix 1 is now added as Supplementary material

- The order in Results is changed: now we report about the literature review first

 - we added additional literature and improved the English language

- Comment: “The study written by Nina Šajna and colleagues focuses on Pistia stratiotes, alien aquatic plant, their occurrence in thermal water and ecological, physiological and biomass characteristics. The manuscript could be interesting for research of the international journal Diversity. However, according to my opinion the manuscript in recent form is a hybrid between review and traditional research papers. Part of the manuscript is focused on review about occurrence of the aquatic plants in thermal waters in Europe and especially, P. stratiotes. Second part showed results of ecological and physiological research of P. stratiotes in specific thermal water in Slovenia done by the authors. I think that the manuscript should be divided into two single manuscripts. The first, review about alien aquatic plants of thermal waters in Europe and the second, results of the research of P. stratiotes in Slovenia. However, such a review paper would need much more work than a research paper. Therefore, the manuscript should be re-written and subsequently again submitted into Diversity journal, but only its “research” part; review paper needs greater work input. Introduction could be changed and more focused on targed species, P. stratiotes. The review about P. stratiotes in Europe, now presented in the Results chapter, could be redirected to Methods chapter (e.g. within subchapter “Studied species”) and the Discussion chapter. Review part about other alien aquatic plants should be omitted, similarly as the part of the Discussion chapter which focuses on the European alien aquatic plants.”

The paper was submitted to the special issue “Diversity of Inland Wetlands: Important Roles in Mitigation of Human Impacts.” We wanted to report that some inland waters are thermally abnormal and can provide safe sites for tropical/subtropical plants that are problematic now and in the future; in thermally abnormal and normal waters. The paper consists of two related parts. The first part comprises results obtained by a literature review as a method to evaluate the diversity of alien plants in thermally abnormal waters, which enabled us to pinpoint that P. stratiotes is very common, even beyond thermally abnormal waters; second the focus on P. stratiotes in thermal stream in Slovenia. We changed the order in Results and now report first about the plants in thermally abnormal waters from literature and then about P. stratiotes. We put Table 1 in an appendix to gain more clarity. We added additional literature, also the proposed paper Ružičková et al. 2020. Original data is now in supplementary file. We believe that literature data on plants in thermally abnormal waters is important and makes the paper suitable for broader readership, if we don’t focus only on the presented case study of P. stratiotes. The paper also includes large original dataset, which might be used for future comparisons. The decision on the paper was major revision. 

Sincerely,

the authors.

Reviewer 3 Report

How fast do these plants grow? doubling time? 

When are the winter and summer months? This can help in Fig1C, when comparing plant cover in the different months. Would you say the Wastewater treatment plant is contributing to the conditions for the invasive plant? An analysis of the water would help to determine if there are other factors contributing to the growth of the plant, especially before and after the wastewater treatment.

In Fig 2, there are two shades of grey. please indicate dark grey or light grey so as to not confuse.

Normally it is written as 1980s and 1990s not 1980-ies

In the review it would be interest to state what the percentage coverages are of the plant in the various waterbodies to see the extent of the invasion.

English editing is needed as it made reading a little difficult.

It would have been good to indicate what other factors contributed to the growth of the plant in the other regions to see if the thermally abnormal waters were the main factors?

Author Response

Dear reviewer thank you very much for your time and helpful comments improving our paper. Here we first list the major changes we made and later address all the comments in detail:

- Table 1 has moved to Appendix 1

- former Appendix 1 is now added as Supplementary material

- The order in Results is changed: now we report about the literature review first

 - we added additional literature and improved the English language

- Comment: “How fast do these plants grow? doubling time?”

Evaluating a single plant’s growth is somewhat difficult for P. stratiotes. The “mother” plant grows and forms “daughter” plants on stolons which break off easily. The doubling time of the mother plant depends on how much biomass is “lost” on the daughter plants. The number of daughter plants depends on the density of all plants. Mother plants from which many daughter plants break off are weaker competitors. There is no published data on how biomass allocation/nutrients between mother and daughter plants is happening. We also did not find data in the literature for biomass doubling time for P. stratiotes; it is known for Eichornia crassipes to be less than 1 week (Hussner et al. 2021), though.

- Comment: “When are the winter and summer months? This can help in Fig1C, when comparing plant cover in the different months.”

The duration of summer and winter months is now included in the text (p. 13?)

- Comment: “Would you say the Wastewater treatment plant is contributing to the conditions for the invasive plant? An analysis of the water would help to determine if there are other factors contributing to the growth of the plant, especially before and after the wastewater treatment.”

Data about phosphates and nitrates measured during our study is in the Supplementary file (former Appendix 1). Nitrates showed seasonal variability – they were high in March and April when nitrates also increased downstream. However, values did not increase significantly between before (sampling point 5) and after the wastewater treatment plant (sampling point 6). Phosphates did not show any pattern downstream nor the difference between sampling points 5 and 6. We used colorimetric kits and these might be not very sensible. In our first study (Sajna et al. 2007), when we used detailed chemical analysis, upper sampling points showed about 60% fewer nitrates than lower points in December. Later in the season, nitrates rose along the entire stream. Ammonia–nitrogen and orthophosphate did not vary greatly between sampling sites year-round but increased in summer not depending on the position of the water treatment plant. Therefore, we concluded that in our study site temperature is more important than nitrogen and phosphorus, even though nutrients are the primary factor determining growth.

- Comment: “In Fig 2, there are two shades of grey. please indicate dark grey or light grey so as to not confuse.”

Thank you. We added a description of the meaning of different grey colors in the figure caption.

- Comment: “Normally it is written as 1980s and 1990s not 1980-ies”

Thank you. Corrected in the entire paper.

- Comment: “In the review it would be interest to state what the percentage coverages are of the plant in the various waterbodies to see the extent of the invasion.”

We agree. Unfortunately, data on cover evaluated in total area or in e.g. m2 is scarce. For P. stratiotes, it often covers the entire surface of a waterbody, studies frequently report biomass (e.g. mass per m2) instead of cover.

- Comment: “English editing is needed as it made reading a little difficult.”

We corrected the English language – spelling mistakes and improved language style.

- Comment: “It would have been good to indicate what other factors contributed to the growth of the plant in the other regions to see if the thermally abnormal waters were the main factors?"

Nutrients are now mentioned in the line 345.

Sincerely, the authors

Round 2

Reviewer 2 Report

Dear authors, thank you very much for answers on my notes in the first version of my review. I think that the new version of the manuscript is the appropriate to publish in the Diversity journal with two small changes.

189, " no reports" better "no summary overviews"

Appendix 1. Please add reference "http://sbs.sav.sk/SBS1/bulletins/docs/bulletin42_2/BSBS-2020-2_ZFN.pdf" for Rotala rotundifolia; the species was recently published from the territory of Slovakia from thermal waters

Author Response

Dear reviewer thank you for your help. We corrected the proposed formulation "no summary overview" and added the suggested reference for R. rotundifolia.